# Experimental Study on the Development of Fly Ash Foam Concrete Containing Phase Change Materials (PCMs)

**DOI:** 10.3390/ma15238428

**Published:** 2022-11-26

**Authors:** Purev-Erdene Bat-Erdene, Sanjay Pareek

**Affiliations:** Department of Architecture, College of Engineering, Nihon University, Koriyama 963-8642, Fukushima, Japan

**Keywords:** phase change material (PCM), thermal energy storage, heating and cooling load

## Abstract

Phase change materials (PCMs) have the ability to absorb and release a large amount of energy during the process of transforming physical properties (i.e., phase transition process). PCMs are suitable for thermal energy storage and reducing energy consumption in buildings. The aim of the study is to assess the basic material properties and thermal behavior of fly ash foam concrete mixed with two different types of microencapsulated PCMs (PCM6D and PCM18D). We made five different varieties of fly ash foam concrete by replacing the equivalent unit weight of cement with PCM 0%, PCM 10% and PCM 30%. The results show that using a new type of mixer, the microencapsulated PCMs kept their spherical shapes without any cracks or damage in the foam concrete matrix. Differential scanning calorimetry analysis showed that PCM18D-30% had a latent heat capacity of 19.2 °C and 44.7 J/g, in liquid and solid phase with melting and freezing temperatures of 9.46 °C and 41.7 J/g respectively. Additionally, thermocycle analysis showed that it had maintained the temperature for 8 h within the phase change range. In conclusion, PCMs can reduce indoor temperature fluctuations and exhibit the potential for enhancing energy savings and thermal comfort of buildings.

## 1. Introduction

In many countries of the world, energy consumption has increased exponentially. Energy use in construction accounts for almost a third of the total consumption in most countries [1]. That is largely due to the increase in the living standards and comfort demands, especially for heating and cooling purposes. For example, an average 64.1% of energy is used to heat households, according to studies conducted in the residential sector in Europe [2]. Current indoor heating is increasingly dependent on active systems to ensure indoor thermal comfort, which is non-sustainable in the long therm. Thus, it is necessary to make efforts to reduce the energy consumption in residential buildings.

An effective solution to this increasing problem is the use of renewable energy in the building sector [3,4,5]. Taking advantage of solar thermal energy can help reduce heating and cooling energy consumption. To do so, the use of phase change materials (PCMs) is proposed due their capacity to store a high amount of thermal energy. In addition, the alternating sequence process of the melting and solidifying of PCMs functions as a heating and cooling system. Energy is absorbed when the temperature around PCM increases, and it changes its phase from solid to liquid. Conversely, with a temperature drop, PCM freezes by changing its phase into solid state [6].

In the building sector, it is suggested that this phase changing feature could offer various applications. For instance, this material can be used for walls, roofs and floors. In order to integrate PCMs into the building materials, two methods are commonly used: direct impregnation and microencapsulation of PCMs. In the first method, PCMs are absorbed into porous aggregates to form stable composite, which allows for use in construction materials [7,8,9,10,11]. However, while absorbing PCM into the porous particles, PCM may leak out of the porous material during melting because there is no protective layer on the surface of the composite material, which leads to reduced thermal storage capacity. As a result, the direct impregnation method can have a negative impact on thermal storage capacity [12].

In the second method, PCMs are microencapsulated within a protective shell, with a particle size between 15 and 30 μm [13]. This protective shell allows microencapsulated PCMs to be preserved during heating and cooling cycles. Recent studies have shown that microencapsulated PCMs can have a positive effect on thermal performance when incorporated with porous material [13,14,15,16]. However, microencapsulated PCM has several drawbacks when used extensively. PCM’s shell protection is made of polymers with low mechanical stiffness and strength. As a result, when integrating microencapsulated PCM into construction material, mechanical strength can be significantly reduced [17,18,19,20]. Because of low mechanical strength, PCM microcapsules can also be damaged during the mixing process when they are directly integrated into a cementitious composite [18,20]. This process leads to significant loss of energy storage potential.

In order to make PCM more widely practical as a building material, the above barriers must be removed. This issue can be addressed by using a suitable mixer, adjusting the speed and mixing with lightweight materials such as foam concrete. The use of foam is expected to prevent cracking of the PCM shells during the mixing process. Foamed concrete generally refers to lightweight concrete with a significantly reduced density, high porosity and low thermal conductivity [21,22,23,24]. Due to its high volume, air voids and light weight, it can be used as an insulation material for external walls [24,25].

Therefore, it is important to test the possibility of incorporating microencapsulated PCM without any damage into fly ash foam concrete, which can melt and absorb heat during the daytime when the outside temperature increases. Then, the absorbed energy is kept at constant temperature. Later, when the temperature decreases, the phase changes from a liquid to a solid state, releasing all of the energy absorbed back into the environment [26], as shown in Figure 1. As a result, concrete materials incorporated with PCMs can provide a more stable temperature conducive to human thermal comfort in buildings. Developing such a new type of fly ash foam concrete that incorporates several kinds of PCM with varying temperature ranges by taking advantage of their specific characteristics is the primary aim of this research. The results further confirmed the energy saving potential of the PCM fly ash foam concrete, and consequently, provided the groundwork for better understanding the phase change phenomena and experimental approaches.

## 2. Materials and Method

### 2.1. Materials

Two types of microencapsulated PCM were studied: PCM6D (melting temperature: 4 to 8 °C) and PCM18D (melting temperature: 16 to 20 °C). The properties of the PCMs, provided by the manufacturer, are presented in Table 1 and Figure 2. Fly ash (type 2 fly ash specified in Table 2) and rapid hardening cement (RHC-Table 3) were used. A water reducing admixture (Mighty 3000H) was used to improve the workability of PCM foam concrete. The foaming agent (Drexel) was also used as its foam stability and steady properties intensify the structure formation of foam concrete when added to cement mortar.

### 2.2. Mix Proportion

The test samples were 40 mm × 40 mm × 160 mm in size, and PCM was replaced in ratios of 0%, 10% and 30% (cement ratio) by both PCMs (PCM6D and PCM18D) in 3 samples, respectively, and a total of 5 samples were prepared (Table 4). Previous researchers reported that PCM tends to be easily damaged during the mixing process when it is directly integrated into a cementitious composite [18,26]. To not crush the PCM, an OM mixer was used in the fly ash foam concrete. It is a device that does not have a stirring blade, but has a flexible rubber ball mounted on a rocking board, which can help the PCM to mix homogeneously, thus preventing any damage to PCM particles in the concrete mixture as shown as Figure 3. First, the dry materials such as cement, fly ash and PCM were mixed thoroughly. Next, water and a high-performance water reducing agent were added, and were mixed uniformly. Afterwards, foam was produced by introducing compressed air at 0.4 MPa through the nozzle of the foam generator as shown as Figure 4. Prior to the addition of foam to the mortar, the density of mortar was determined by pouring the mortar into a known volume-measuring container to measure its weight (Figure 5). All of the mix composition was calibrated in order to achieve target densities. Water, cement and the foam content were adjusted to obtain the target wet density. The mixture was cured in a 3D wet chamber for 3 d [20 °C, 90% (RH)], and then in a constant temperature and humidity chamber [80 °C, 90% (RH)]. The total curing periods between tests were 3 d, 7 d and 28 d.

### 2.3. Mechanical Properties

The compressive strength and density tests were performed at 28 days of age according to [29] standard. The samples were extracted in triplicate and for every composition that was tested, results reported were the average of the three samples.

### 2.4. Thermal Analysis

#### 2.4.1. Thermal Properties Analysis (DSC)

In order to evaluate the energy storage capacity, Differential scanning calorimetry (DSC) was used. The DSC was taken with a TA device with a temperature accuracy of ± 0.1 °C. It measures the difference in the heat flow rate between samples and inserts reference as a function of time and temperature [30]. The melting and freezing behavior of the PCMs were analyzed using a DSC model Q200. The temperature was measured at the sample with a thermocouple. The sample sizes were between 5 and 10 mg and were encapsulated with pans of the sample material (and the lid cap of the container) [31]. The samples were weighed using an analytical ultra-micro balance model: MC25. Then the samples were sealed in pans the using an encapsulating press. An empty aluminum pan was used as a reference for all measurements.

In order to optimize the experimental parameter governing the DSC test, many investigations were analyzed using various dynamic rates and samples. In this way, different heating and cooling rates of 0.5 °C min^−1^, 1 °C min^−1^, 2 °C min^−1^, 5 °C min^−1^ and 10 °C min^−1^ were considered. The program of the test procedure is as follows: Starting with an initial isothermal period at 20 °C for 1 min prior to dynamic heating, up to +40 °C according to the proposed rates (0.5 °C min^−1^, 1 °C min^−1^, 2 °C min^−1^, 5 °C min^−1^ and 10 °C min^−1^).

#### 2.4.2. Thermal Conductivity

Four 200 mm × 200 mm × 10 mm samples for each type were used to characterize the thermal conductivity according to [32] standard. Thermal conductivity of all the specimens was measured after 7 days of drying. The measuring instrument was the Fox-200. The optimal delta of temperature between the plates (lower and upper) suggested by manufacture is 20 °C [33,34], according to [32] standard for steady-state measurements, and the arithmetic mean of temperature (T1 + T2)/2 was adopted as the repetitive temperature for measuring thermal conductivity.

#### 2.4.3. Experimental Program of Heat Cycle Test

The experimental program of the heat cycling test is shown in Figure 6 [35]; in each thermal 21 h cycle, the chamber temperature was maintained at 10 °C for 4 h (at rate of 2.5 °C/min), then the chamber temperature was heated from 10 °C to 50 °C in 20 min and maintained at 50 °C for 8 h, then it was cooled to 10 °C in 20 min and maintained at 10 °C for 8 h.

#### 2.4.4. Preparation of Heat Cyclic Test Samples

The thermal cycle test was carried out by placing the sample in a chamber with a programmable humidity and temperature (model ETAC dc-450) Figure 7. In this research, 2 types of PCM foam concrete were studied

In order to assess the effect of PCM foam concrete used as a wall for buildings, two closed prototypes were built with laboratory-scale dimensions. The first prototype was measured from inside to outside: a 40 mm thick layer of PCM foam concrete around 200 mm × 200 mm × 20 mm specimen, and a 50 mm thick extruded polystyrene foam around 300 mm × 300 mm × 300 mm specimen. The schematic diagram of the physical models and cross-section of the model are shown in Figure 8. In this prototype, three thermocouples were used (Figure 7 and Figure 8). One was put on the backside of the specimen, the other at the surface of the specimen. Another one was installed in the middle of the specimen. The thermocouples in the middle of the specimen were used to compare with different sides of the thermocouple.

## 3. Experimental Results and Discussion

### 3.1. Compressive Strength Test

Figure 9 shows the compressive strength test results of foam concrete with and without PCM. The tests were conducted under the curing temperature of 40 °C, RH 95%. Two types of PCMs were used, which were PCM6D and PCM18D. As seen from Figure 9, PCMs were added to replace 10% and 30% of unit weight of cement in the foam concrete. On day 7, the compressive strength of PCM6D-10% and PCM18D-10% were recorded 5.6 MPa and 7.3 MPa, respectively, whereas foam concrete without PCM revealed strength of 9.35 MPa after 7 days of curing. The decrease in the strength of PCM6D-10% and PCM18D-10% concrete compared with that without PCM was around 40.1% and 21.9% respectively, after replacing cement by 10%. Dramatic change was observed on day 28. PCM6D-30% and PCM18D-30% were noticed to have strengths of 3.5 MPa and 4.5 MPa, respectively, whereas No-PCM concrete revealed a strength of 13 MPa after 28 days of curing. The decrease in strength of PCM6D-30% and PCM18D-30% concrete compared with those without PCM was around 73.1% and 66.9% respectively, compared with the results of with those of without PCM, after replacing 30%. On the other hand, the compressive strength increased from day 7 to day 28; the percentages were for foam concrete without PCM: 41.9%; PCM6D-10%: 20.2%; PCM18D-10%: 19.2%; PCM6D-30%: 25.7% and PCM18D-30%: 41.6%, respectively.

The reduction of compressive strength can be explained as follows: From the results, it can be observed that as the amount PCM increased, the foam volume also increased. The high amount of foam causes decreasing strength of PCM foam concrete. The additive PCMs do not seem to contribute to the compressive strength. They are more like voids than aggregates. The results are in line with the conclusion of [36], which reported a delay in the hydration process of cement mortars containing PCM. It is a well-known fact that PCM does interact with C-S-H formation, thus reduces strength [37,38,39]. Although strength was dramatically decreased, it is sufficient strength to use as an ALC, which is known as an autoclaved aerated concrete panel.

### 3.2. Density Test

For the foam concrete, consistency and stability are the main characteristics of the density test. It can be seen in Table 5 that the stability of foam concrete is the ratio of fresh density to hardened density. As for the consistency, it is the ratio of fresh density to the designated density. Consistency depends on the amount of foam added to the foam concrete mix. Overall, the density of foam concrete depends on the stability and consistency of the mix that translates to the strength of the foam concrete.

Our results show that all the PCM mix compositions had similar consistency and stability that were close to the target densities. Additionally, the foam was able to mix with foam concrete thoroughly without collapsing when the OM mixer was used.

Figure 10 shows the test results of PCM6D and PCM18D concrete density that had 10% and 30% of PCM on days 3, 7, 14 and 28. The control wet density of all of PCM foam concretes was 1100 kg/m^3^. It is clear that the value of PCM6D and PCM18D foam concrete density ranges between 1086–730 kg/m^3^. The highest density was observed at PCM18D-10% (1086 kg/m^3^) on the third day while the lowest density was for PCM6D-30% (730 kg/m^3^) on day 28. There is a density drop from day 3 to day 28, and the percentages were calculated as different as follows: for concrete PCM6-10%: 8.7%; PCM6D-30%: 21.3%; PCM18D-10%: 7.9% and PCM18D-30%: 13.6%, respectively. The results revealed that as the amount of PCM increased, the density of foam concrete decreased. It is because of the relatively low specific gravity of PCM when compared to other components in the foam concrete [30].

### 3.3. DSC and Thermal Analysis

The DSC test on PCM6D and PCM18D foam concretes was conducted by means of five different heating/cooling rates of 10 °C/min, 5 °C/min, 2 °C/min, 1 °C/min and 0.5 °C/min. The results of heating and cooling cycles on foam concrete PCM6D (10% and 30%) samples are shown in Figure 11 and Figure 12. 

When scrutinizing all scanning rates of PCM6D foam concretes, the lowest latent heat (Δh) was found at PCM6D-10%, offering the heating/cooling rates of 0.5 °C/min. According to the graphs in Figure 11, the first peaks of those two were found at 4.75 °C and −2.74 °C, and the melting and freezing latent heats (Δh) stored in unit weight were 3.02 J/g and 3.30 J/g. The second peaks were found at 26.6 °C and 23.7 °C, and the melting and freezing latent heats (Δh) stored in unit weight were 4.44 J/g and 3.17 J/g. Additionally, the highest latent heat (Δh) was found at PCM6D-30%, with heating/cooling rates of 10 °C/min (Figure 12). The melting and freezing temperatures are shown with two peaks in the line graphs. The first peaks of those two were found at 5.99 °C and −4.14 °C, and the melting and freezing latent heats (Δh) stored in unit weight were 13.4 J/g and 12.3 J/g. The second peaks were found at 27.5 °C and 22.1 °C, and the melting and freezing latent heats (Δh) stored in unit weight were 21.4 J/g and 22.8 J/g.

The results of heating and cooling cycles on foam concrete PCM18D-10% and PCM18D-30% samples are shown in Figure 13 and Figure 14. When scrutinizing all the scanning rates of PCM18D foam concretes, the highest latent heat (Δh) was found for PCM18D-30%, at heating/cooling rates of 10 °C/min. The melting and freezing temperatures were shown with one peak in the line graphs. The peaks of those two were found at 19.2 °C and 9.46 °C, and the melting and freezing latent heat (Δh) stored in unit weight were 44.7 J/g and 41.7 J/g. Additionally, the lowest latent heat (Δh) was found at PCM18D-10%, offering the heating/cooling rates of 0.5 °C/min. According to the graphs, the peaks of those two were found at 16.9 °C and 11.0 °C, and the melting and freezing latent heat (Δh) stored in unit weight were 10.3 J/g and 9.3 J/g. Overall, DSC analysis shows that latent heat increased with amount of PCM. The DSC results of PCM6D and PCM18D foam concrete for all scanning rates are summarized in Table 6.

#### Effects of the Heating and Cooling Process on the Results of DSC

Table 6 below summarizes the DSC results of the samples for heating and cooling processes. The DSC curves of the samples PCM6D-10%, PCM6D-30%, PCM18D-10% and PCM18D-30% are shown in Figure 15 and Figure 16 to illustrate their heating and cooling processes. By focusing on the start and end of the phase change, the onset and end temperatures were identified from the intersections of the inflection points of PCM6D-10%, PCM6D-30%, PCM18D-10% and PCM18D-30%, respectively. How the onset and end temperature were identified is shown below (Figure 15) using PCM6D-10% as an example. It was recorded that the melting process started at about −6.40 °C and ended at 9.5 °C during the heating. During the 10 °C/min range, the heat was removed from the sample and the freezing started at around −1.31 °C and ended at −11.26 °C.

The Figure 15 and Figure 16 compare the onset, end and peak temperatures of all PCM types. As for the comparison between PCM6D-10% and PCM6D-30%, it is clear that there were small differences between the heating/cooling rates, with an almost negligible behavior observed. Such an example is presented in Figure 15, which focuses on the PCM6D-10% and PCM6D-30% samples, where a negligible shift can be seen between the rates of 0.5 °C/min and 10 °C/min. Unlike PCM6D, the comparison of the onset, end and peak temperatures of PCM18D-10% with those of PCM 18D-30% (Figure 16) reveals that there were some differences between the heating rates, but there was only a negligible difference in behavior observed for the cooling rates. It is apparent that the fluxes were normalized to their peaks so that simultaneous observation of the thermogram shape for 0.5 °C/min and 10 °C/min was facilitated. In general, it can be seen that the PCM peak response is inclined to the direction of imposed flux, which means higher heats for heating and lower peak temperatures for cooling. Another noteworthy fact is that PCM works within that range, meaning that there were no breaks observed in the PCMs in the concrete matrix.

### 3.4. Thermal Conductivity

The effect of the replacement of PCM into foam concrete specimens in thermal conductivity was measured on day 28. From PCM6D-10% to PCM18D-30%, each designated mix used to measure the thermal conductivity was plotted in Figure 17. Thermal conductivity of PCM foam concrete decreased with the increasing amount of PCM substitution. Compared to the reference, the reduction percentages of thermal conductivity for the composite PCM6D substitution levels of 10% and 30% were found at 45.8% and 52.2%, respectively.

Moving to PCM18D, substitution levels of 10% and 30% were found at 21.4% and 55.3%, respectively. The reduced trend of thermal conductivity with the increasing substitution level of PCM composite is attributed to the low thermal conductivity from the PCM6D to PCM18D composites (in the range of 0.1–0.3 W/m·K).

The reason for this can be the impact of PCM substitution, and the fact that independent cell foam does not allow heat loss from the pores, which may affect lower conductivity values [25,40].

### 3.5. Thermocycle Analysis

PCM6D mainly has two melting ranges, the first one was 4–8 °C and the second one 18–24 °C. The temperature behavior ranges from 10–50 °C. The cycling conditions were a total of 24 h per cycle through heating and cooling for 8 h with a heating rate of 2.5 °C/min. From heat cycle tests of PCM6D-10% and PCM6D-30% foam concretes, the lowest latent heat was found for PCM6D-10%. There was almost no difference compared to No-PCM foam concrete. The highest latent heat was found in PCM6D-30% (Figure 18). Considering the PCM6D-30%, when the temperatures increased from 4 °C to 24 °C, PCM foam concrete starts melting and delays the temperature for around 1 h. In contrast, when the temperature decreased from 24 °C to 4 °C, PCM foam concrete starts freezing. The temperature was delayed for around 1 and a half hours. The melting range of PCM18D was 16–20 °C. The results of heating and cooling cycles on foam concrete PCM18D samples are shown in Figure 19. When we heat cycle tested PCM18D-10% and PCM18D-30% foam concretes, the lowest latent heat was found in PCM6D-10%. Considering the PCM18D-10%, when the temperatures increased from 10 °C to 20 °C, PCM foam concrete started melting and delayed the temperature for around 1 h.

Contrarily, when the temperature decreased from 20 °C to 10 °C, PCM foam concrete started freezing. The temperature was delayed for around 3 h. The highest latent heat was found in PCM18D-30%. Considering the PCM18D-30%, the temperatures increased from 10 °C to 20 °C, PCM foam concrete started melting and delayed the temperature for around 3 h. Contrarily, when the temperature decreased from 20 °C to 10 °C, PCM foam concrete started freezing. The temperature was delayed for around 8 h.

### 3.6. Microstructure Analysis

The observation of the microstructure of the PCM foam concrete mix by SEM produced the images of the PCM foam concrete samples. These contain (A) No-PCM, (B) PCM-10% and (C) PCM-30%, which are shown in Figure 20. The same effects can be seen for both PCM6D and PCM18D, that from the observation both fly ash particles and PCM capsules are dispersed in the foam concrete matrix. It is said that the cementitious matrix that is composed predominantly of calcium-silica hydrate (C-S-H) and portlandite (CH) host the inert inclusion of quartz [37]. Ref. [41] said that these two phases are the main products of cement hydration. During the mixing process, the microcapsule PCMs kept their spherical shapes without any cracks or damage.

## 4. Conclusions

This research presents an overview of the influences of the basic material properties and thermal behavior of foam concrete mixed with PCM to identify the contribution of PCMs to the mixture. From the results, the following conclusion can be made:As a result of mechanical tests, the 28 day compressive strength of both PCM6D-30% and PCM-18D-30% were 3.5 MPa and 4.5 MPa, respectively. Both types of fly ash foam concrete have sufficient mechanical properties to be used in applications such as the ALC panel with compressive strength of 3 MPa.The results of the DSC test showed that it indicates that all types of PCM foam concrete performed as predicted, melting and freezing within the exact temperature rates and the energy storage capacity increased with amount of PCM in the foam concrete. PCM18D-30% had a latent heat capacity of 19.2 °C and 44.7 J/g in liquid and solid phase with melting and freezing temperatures of 9.4 °C and 41.7 J/g, which was highest of all other types.The results of thermal conductivity showed that the increasing substitution level of PCM composite attributes to the low thermal conductivity from the PCM6D and PCM18D composites (in the range of 0.1–0.3 W/m.K).The results of the thermocycle analysis found that PCM foam concrete types performed well, maintaining the temperature within the range for a certain period of time. The highest maintained temperature was found for PCM18D, for which the temperature delay was the longest, around 8 h.As a result of SEM analysis, the microencapsulate PCM kept their spherical shapes without virtually any cracks or damage. The OM mixer was shown to be highly effective for mixing PCMs in the foam concrete, which is one of the fundamental findings of this research.

To conclude, PCM foam concrete had some useful characteristics such as better thermal performance and latent heat storage. The results revealed that PCMs can reduce indoor temperature fluctuations as they exhibit the potential for enhancing energy savings and thermal comfort of buildings. Moreover, future studies can use the energy simulation program to provide comparative results for PCM foam concrete, and, based on the experimental and simulation data, it can be further investigated at the real scale dimension of buildings.

## Figures and Tables

**Figure 1 materials-15-08428-f001:**
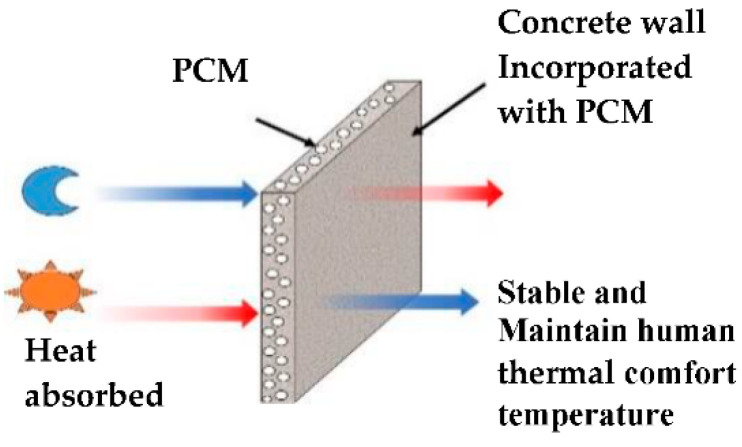
Structure and working principle of PCM as a building wall.

**Figure 2 materials-15-08428-f002:**
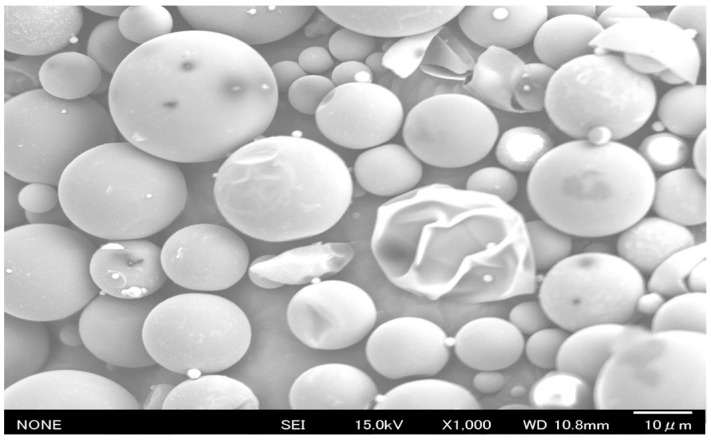
SEM image of pure PCM6D (×1000).

**Figure 3 materials-15-08428-f003:**
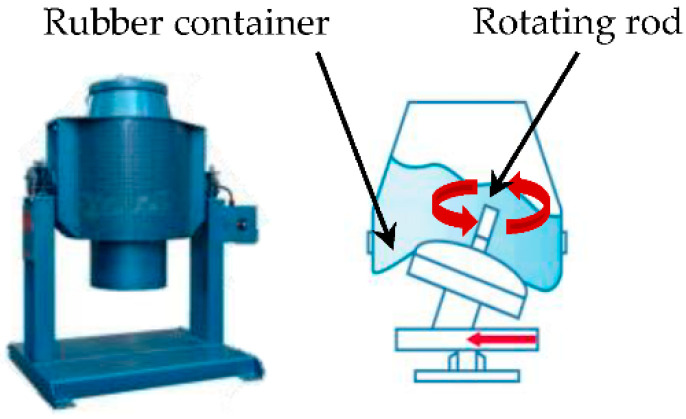
OM mixer and operation [28].

**Figure 4 materials-15-08428-f004:**
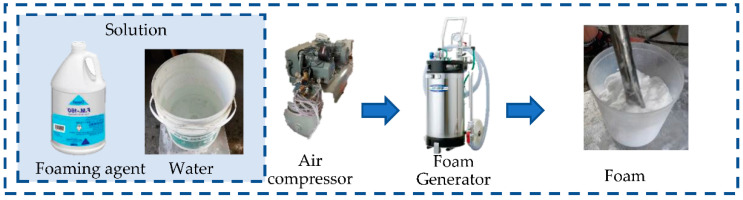
Procedure of making foam.

**Figure 5 materials-15-08428-f005:**
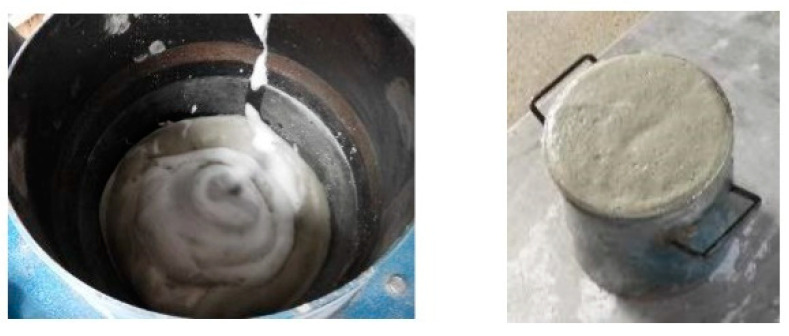
Mixing process and foamed concrete.

**Figure 6 materials-15-08428-f006:**
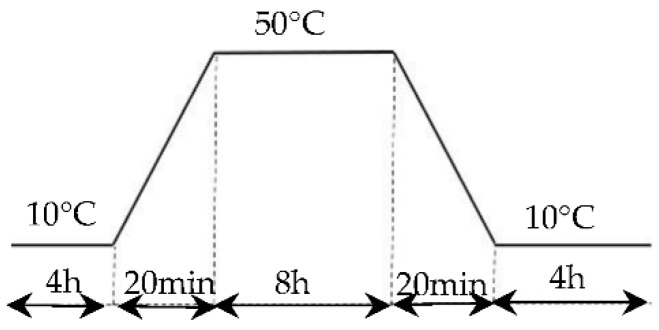
Heat cycling program.

**Figure 7 materials-15-08428-f007:**
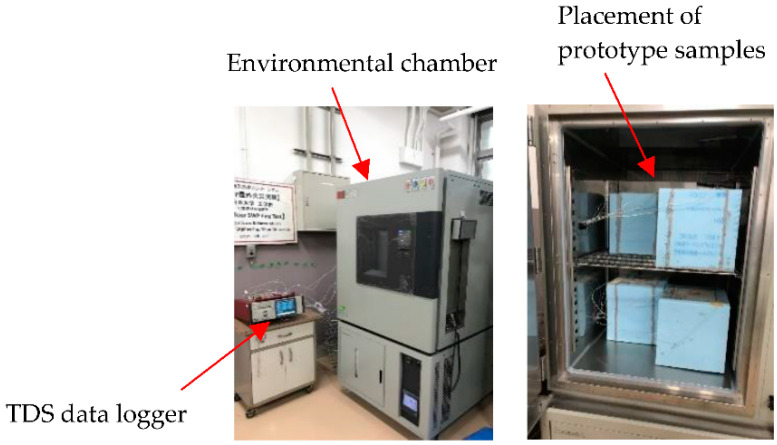
Environmental chamber for heat cyclic test.

**Figure 8 materials-15-08428-f008:**
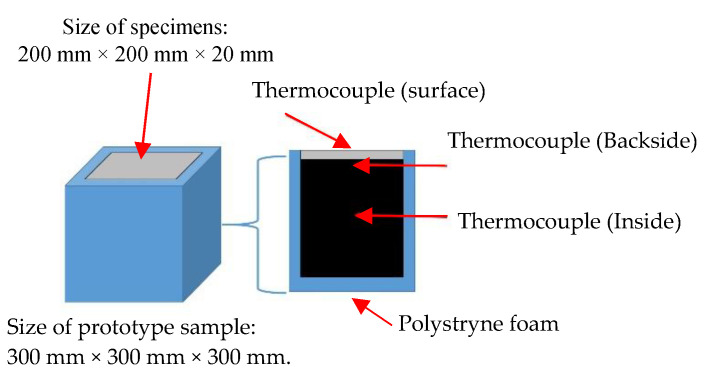
Prototype sample for heat cyclic test.

**Figure 9 materials-15-08428-f009:**
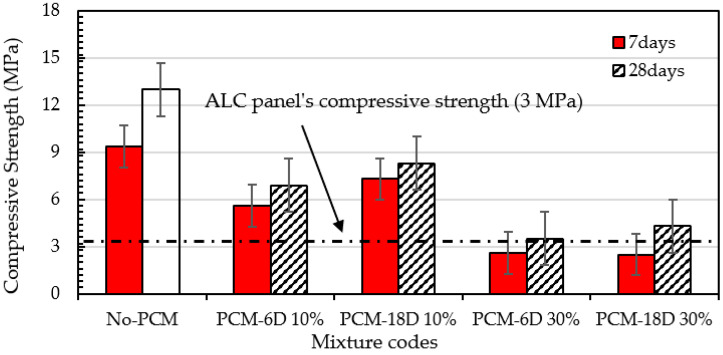
Effect of adding PCM on compressive strength of foam concrete.

**Figure 10 materials-15-08428-f010:**
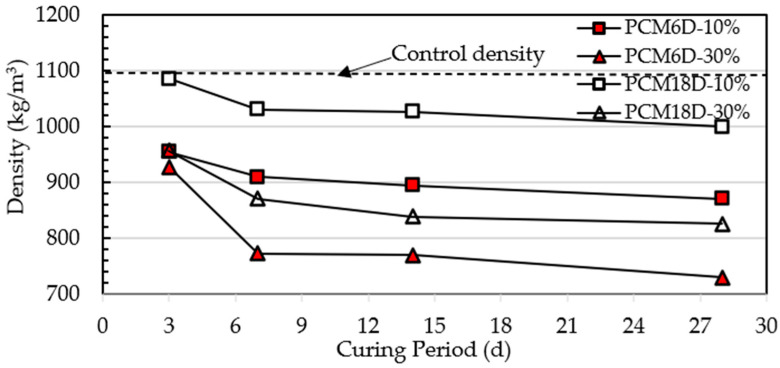
Density of the foam concrete after adding PCMs.

**Figure 11 materials-15-08428-f011:**
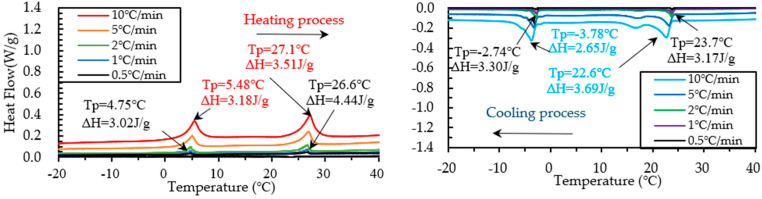
Results of DSC measurement of PCM6D-10% at various heating and cooling rates.

**Figure 12 materials-15-08428-f012:**
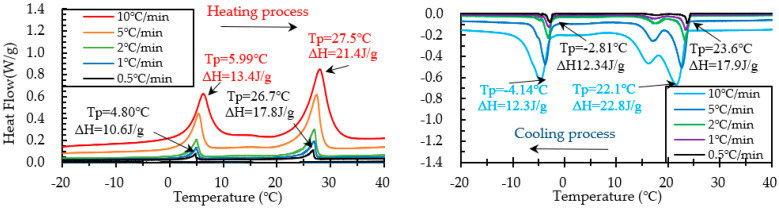
Results of DSC measurement of PCM6D-30% at various heating and cooling rates.

**Figure 13 materials-15-08428-f013:**
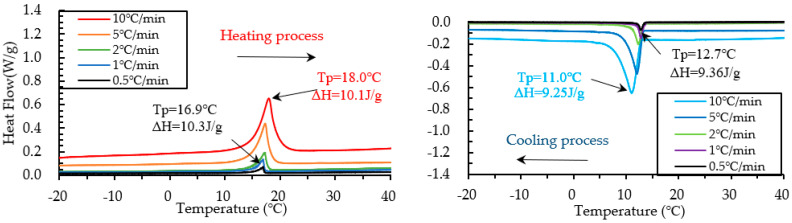
Results of DSC measurement of PCM18D-10% at various heating and cooling rates.

**Figure 14 materials-15-08428-f014:**
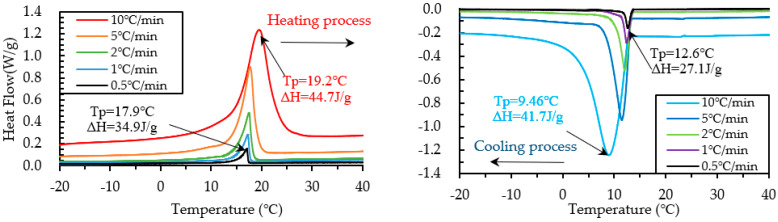
Results of DSC measurement of PCM18D-30% at various heating and cooling rates.

**Figure 15 materials-15-08428-f015:**
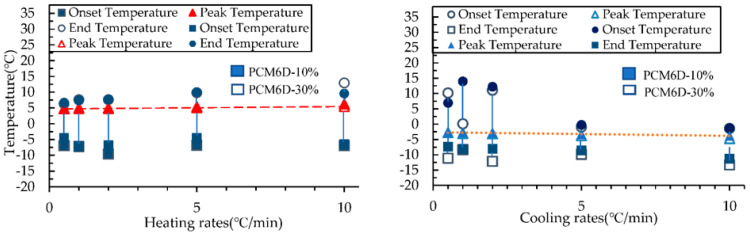
Onset, peak and end temperature for the phase transition in PCM6D heating and cooling rates.

**Figure 16 materials-15-08428-f016:**
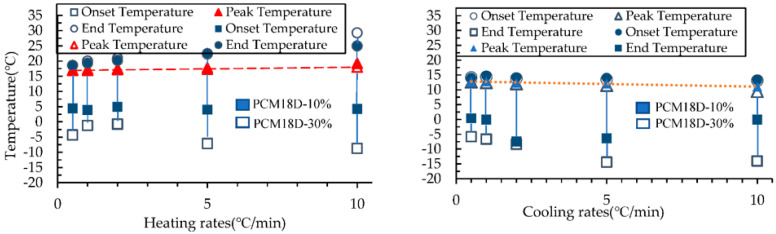
Onset, peak and end temperature for the phase transition in PCM18D heating and cooling rates.

**Figure 17 materials-15-08428-f017:**
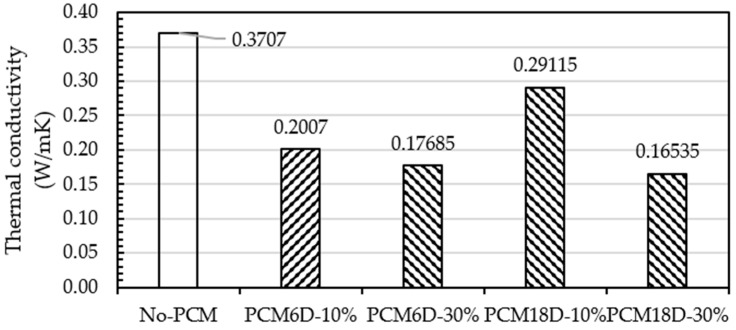
Thermal conductivity test results of foam concrete with and without PCM.

**Figure 18 materials-15-08428-f018:**
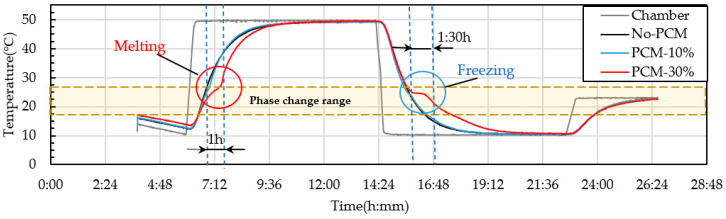
Experimental condition for heat cyclic test (different amounts of PCM6D).

**Figure 19 materials-15-08428-f019:**
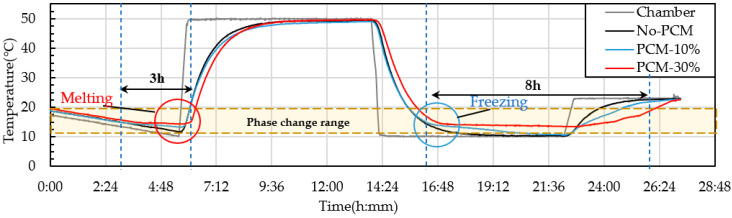
Experimental condition for heat cyclic test (different amounts of PCM18D).

**Figure 20 materials-15-08428-f020:**
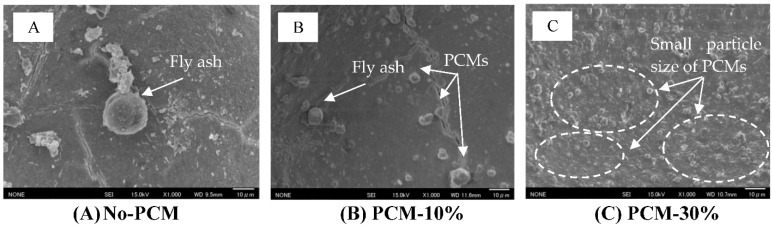
SEM image of PCM foam concrete after 28 days of curing (×1000): (**A**) No-PCM, magnification ×1000 (**B**) PCM6D-10% magnification ×1000 (**C**) PCM6D-30%, magnification ×1000.

**Table 1 materials-15-08428-t001:** Physical properties of different encapsulated PCMs studied.

Material	Nature	Size μm	Melting (°C)	Heat of Fusion (J/g)	Solid Content %
PCM6D	Dry	15.0–30.0	4 to 8	181	97.0–100.0
PCM18D	Dry	15.0–30.0	16 to 20	215	97.0–100.0

**Table 2 materials-15-08428-t002:** Chemical composition of fly ash.

	SIO_2_	Al_2_O_3_	Fe_2_O_3_	CaO	MgO	Glass	p (g/cm^3^)	Ig.loss
Japan FA-II [27]	64.5	23.9	4.8	5.3	1.5	77.1	3.14	2.10

**Table 3 materials-15-08428-t003:** Chemical composition of rapid hardening cement.

	SO_3_	Cl	Na_2_Oeq	MgO	C_3_S	C_2_S	C_3_A	C_4_AF	Ig.loss
RHC	2.99	0.007	0.44	1.18	64	12	8	8	1.05

**Table 4 materials-15-08428-t004:** Mix proportion for foam concrete with PCM.

Mix Designation	Cement (kg)	Fly Ash (wt.%) 10%	W/C 0.3	PCM (wt.%) 10–30%	SP (kg) 1%	Foam (g)
No-PCM (0%)	4.50	0.45	1.35	0	0.045	320
PCM 6D (10%)	3.15	0.45	0.94	0.45	0.045	450
PCM 6D (30%)	2.25	0.45	0.67	1.35	0.045	900
PCM 18D (10%)	3.15	0.45	0.94	0.45	0.045	420
PCM 18D (30%)	2.25	0.45	0.67	1.35	0.045	890

**Table 5 materials-15-08428-t005:** Results of density test of PCM foam concrete.

(Density Control)							
Mix Designation	(wt.%)	Curing Condition	Curing Period (d)	Fresh Density (Kg/m^3^)	Dry Density (Kg/m^3^)	Consistency (%)	Stability (%)
No-PCM	0	40 °C R.H-95%	3	1044	1021	1.0	1.0
			7	1044	1016	1.0	1.0
			14	1044	1012	1.0	1.0
			28	1044	1008	1.0	1.0
PCM6D-10%	10	40 °C R.H-95%	3	955	955	1.0	1.0
			7	955	910	1.0	1.0
			14	955	895	1.0	1.1
			28	955	871	1.0	1.1
PCM6D-30%	30	40 °C R.H-95%	3	1062	928	1.1	1.1
			7	1062	773	1.1	1.4
			14	1062	770	1.1	1.4
			28	1062	730	1.1	1.5
PCM18D-10%	10	40 °C R.H-95%	3	1003	1086	1.0	0.9
			7	1003	1031	1.0	1.0
			14	1003	1027	1.0	1.0
			28	1003	1000	1.0	1.0
PCM18D-30%	30	40 °C R.H-95%	3	1050	957	1.1	1.1
			7	1050	871	1.1	1.2
			14	1050	839	1.1	1.3
			28	1050	826	1.1	1.3

**Table 6 materials-15-08428-t006:** Different studied encapsulated PCMs with their size, melting point and heat of fusion.

Designation			Melting				Freezing			
			Onset Temp. (°C)	End Temp. (°C)	Peak Enthalpy (J/g)	Peak Temp. (°C)	Onset Temp. (°C)	End Temp. (°C)	Peak Enthalpy (J/g)	Peak Temp. (°C)
PCM6D	10%	10 °C/min	−6.40	9.57	3.19	5.48	−1.31	−11.26	3.69	−3.78
		5 °C/min	−4.55	10.04	3.01	5.05	−0.27	−8.60	3.83	−3.32
		2 °C/min	−6.75	7.72	3.10	4.81	12.24	−8.02	3.65	−2.80
		1 °C/min	−7.21	7.37	3.23	4.80	13.97	−8.14	3.54	−2.74
		0.5 °C/min	−4.55	6.68	3.02	4.75	6.92	−7.36	3.17	−2.74
	30%	10 °C/min	−6.98	12.93	10.60	6.21	−1.42	−13.35	9.35	−4.73
		5 °C/min	−6.86	9.81	11.54	5.36	−0.96	−9.87	9.55	−3.74
		2 °C/min	−9.64	7.61	10.85	4.95	11.06	−12.19	11.25	−3.16
		1 °C/min	−7.21	7.61	11.76	4.94	0.08	−8.37	8.44	−3.03
		0.5 °C/min	−6.98	6.33	10.61	4.8	10.15	−11.15	12.34	−2.81
PCM18D	10%	10 °C/min	4.25	24.97	10.13	18.00	13.28	−0.03	9.36	11.04
		5 °C/min	4.02	22.43	11.46	17.32	14.09	−6.40	10.13	12.06
		2 °C/min	4.94	20.23	9.86	17.22	14.21	−7.44	10.44	12.32
		1 °C/min	3.90	19.30	10.90	16.92	14.78	−0.03	12.06	12.79
		0.5 °C/min	4.48	18.60	10.13	16.92	14.09	0.43	9.25	12.77
	30%	10 °C/min	−8.72	29.26	44.73	19.24	13.16	−14.04	41.76	9.46
		5 °C/min	−7.10	22.31	38.66	17.74	13.51	−14.39	38.28	11.48
		2 °C/min	−0.73	20.92	37.70	17.46	13.86	−8.37	37.24	12.04
		1 °C/min	−1.19	19.99	35.31	17.26	14.44	−6.63	32.49	12.40
		0.5 °C/min	−4.32	18.6	34.91	17.06	14.21	−5.82	27.19	12.60

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
