# Peer review of "Experimental Study on the Development of Fly Ash Foam Concrete Containing Phase Change Materials (PCMs)"

_materials, 2022, doi:10.3390/ma15238428_

Round 1
Reviewer 1 Report
Abstract
A very generic sentences about experimental results have been presented in the abstract. A more relevant sentence about experimental results should be added in the abstract, which explains the tests conducted and results obtained.
Introduction
Please remove Fig.1 and Fig. 2 from the introduction section and provide the numbers in the paragraph form.
Introduction section is very weak. It comprises of only 4 research papers. Please provide a review of about 25-30 papers in the introduction section. One Q1 paper on foam concrete are as follows:
Qasim S. Khan, M. Neaz Sheikh, Timothy J. McCarthy, Mehdi Robati, and Mark Allen (2018)
|
Experimental investigation on foam concrete without and with recycled glass powder: A sustainable solution for future construction. |
Construction and Building Materials, 201, 369-379.
|
Materials and Method
On what basis fly ash is classified as Class type 2? Based on chemical composition provided in Table 2, Fly ash is class F.
Is this the way to cite a standard? JIS A 1108, JSA-JIH H 8453
Experimental Results and Discussions
What is ALC panel?
Fig. 8 is colored and Fig. 9 is black and white, please be consistent.
Define latent heat.
Conclusions
This section is weak.
Provide 4 to 5 clear conclusions supported by the experimental results.
Author Response
Dear Editors,
Thank you very much for giving us an extension to work on the revision of the manuscript. With your kindness, we were able to dramatically improve the manuscript titled “Experimental study on the development of fly ash foam concrete containing phase change materials (PCM)”.
Now the article reads very well, and we revised our manuscript beyond the 4 reviewers’ comments. All the sections had major improvements, for example, abstract is now under 200 words and now it includes the aim of the study, materials methods and major finding with some numbers and figures. Also we removed the figures from the introduction section and now it has a good logic and flow that addressed the necessity of the research and its practical applications. For the results, discussion and conclusion sections, all were revised and more references were provided to reflect the outcome of the study.
With all the present revisions and improvements, we believe that your readers will find our manuscript interesting and easy to read. And therefore, we believe that you’d act favorably in accepting our manuscript for the publication at your well-respected journal.
Yours sincerely,
Bat-Erdene PUREV-ERDENE,

Reviewer 2 Report
I have found the conducted research very interesting, but something remained highly unclear to comprehend and I kindly ask authors to prepare a response letter point-by-point rebuttal and must be subjected to the manuscript as well, considering the following comments with sufficient explanations.
1) What is the main reason that fly ash has been used in this study? Why not the usage of other pozzolanic like silica fume, volcanic ash, slag or the best one like metakaolinite with high reactivity? The reason must be added to the methodology.
2) I am wondering when the temperature drops, PCM freezes by changing its phase into solid state and consequently release energy; therefore, how this released heat transfers to indoor ambient (into the building) not outdoor ambient?
3) Introduction is too short and other highlighted and promising earlier studies must be reported.
4) According to (https://doi.org/10.1680/jcoma.20.00045), Did you also see that the peak density of CSH (tobermorite) to be reduced, and portlandite increases? The literatures concerning CSH contributions are suggested to be added to the introduction.
Author Response

(The authors gave the same response as above.)

Reviewer 3 Report
In this study, the use of 2 different PCMs (PCM6D and PCM18D) in foam concrete was investigated. Although the study is good from an experimental point of view, there are some concerns in writing. The following corrections are required for the study to be published.
Line 46: PCMs should be phase change materials (PCMs).
The writing of the intro is very poor. Some further development is needed by referring to articles in the literature where PCM is used in foam concrete or building materials.
Line 85-86: “The test samples were 40x40x160mm in size, and PCM was replaced with a ratio of 0%, 10% and 30% (mass ratio) by PCMs in 3 samples respectively and a total of 9 samples were prepared.” This sentence does not mention what PCM is used by changing it by mass ratio.
Line 85: It was said that the test samples were 40x40x160 mm in size. However, only compressive strengths are given. It is not mentioned why flexural strengths are not given.
When Figure 5 is examined, it is seen that cycles are made in the temperature range of 10-50 ℃. However, at 10°C, PCM18D is solid, while PCM6D is still liquid. Therefore, it seems unsuitable to work in these temperature ranges. Therefore, I think that either different temperature ranges should be applied for both PCMs or a temperature range should be chosen where both PCMs can undergo solid-liquid phase transition. In addition, no comments regarding this situation were mentioned in the article. This complexity needs to be explained.
“3.1 Compressive strength test”: This section needs to be completely revised. 3-day compressive strengths are mentioned, but in Figure 8, there are 7 and 28 days results. If the 3 days given here are misspelled and will be 7 days, the values given in Figure 8 and the values in the sentence do not match. For example; For the No-PCM coded sample, 7.5 MPa was written, but the 7-days No-PCM coded sample has a compressive strength of approximately 9.5 MPa. Line 168-169: “On day 3, the compressive strength of PCM6D-10% and PCM18D-10% were recorded 5.5 MPa and 6.7 MPa respectively whereas foam concrete without PCM revealed 7.5 MPa after 3 days of curing.”
The X axis of the graph given in Figure 8 should be corrected as "Mixture Codes" instead of "Curing period".
“3.2 Density test”: This section is lacking in discussion. What kind of effect does PCM have on cement based mortars or foam concretes in studies in the literature. How the results of the presented study turned out should be discussed with the literature.
In Figures 6 and 7, it is mentioned that 3 different thermocouples, Backside, Inside and Surface, are used in the setup. To which thermocouple do the temperature measurements given in Figures 17 and 18 belong? I think that the temperatures for 3 different thermocouples should be compared separately. In this way, it can be seen more clearly what kind of temperature change the PCM makes on the sample surface, underneath and inside the box.
“3. 6 Microstructure analysis”: only SEM images for PCM6D are given in this section. Have the same effects been seen for the PCM18D? Even if SEM images are not available, this should be mentioned in the comment section.
“4. Conclusions”: This section should be completely reviewed. I think it would be appropriate to keep it shorter and give it in articles instead of long sentences.
When the article was evaluated in general, the discussion part was weak. Paper writing is very poor.
Author Response

(The authors gave the same response as above.)

Reviewer 4 Report
The study explores the Experimental study on the development of fly ash foam concrete containing phase change materials (PCMs). The manuscript is elaborately described with the help past literatures. All the references cited are relevant to this area of research. The methods are clearly stated with relevant standards. The result and discussion are clearly presented with the help of figures and charts. Conclusion section is supported by the results. However, the following corrections need to be addressed before the acceptance the Manuscript.
1. State the need of using this research in the abstract.
2. Abstract: add your recommendation of research at the last line of the abstract.
3. Key words: It would be better if key wors are arranged in alphabetical order
4. Introduction: Start the introduction section with text. Here, it started with Fig.1 and 2.
5: Mention the novelty of your research.
6. What is the need of using rapid hardening cement?
7. Give the chemical composition of PCM.
8. cite more works and subsequently increase the number of references
9. Present the recommendation/suggestion for future research of your research in the conclusion section.
Author Response

(The authors gave the same response as above.)

Round 2
Reviewer 1 Report
The authors have improved the manuscript. The current manuscript still needs significant improvement. The manuscript lacks clarity. The sequence of tables/figures and explanations do not seem to be matching.
Is their any need to keep photos in the introduction?
Please provide details about the classification of materials such as fly ash etc.
Please provide a picture showing preparation of foam.
Please provide picture of OM mixer used in experiments.
Captions of Tables need to be corrected.
There is alot of confusion among caption numbers of tables and figures.
In graphs/figures major/minor tick marks must be on the inner side and not on the outer side.
Figures 8 to 11, needs to be clearly explained in the manuscript.
Too many lines/explanations in the manuscript, whose meanings are unclear . Lines 219-221, 235-237, 240-242
Author Response
Dear Editors,
We thank you for handling the paper and reviewing it, which helped us to improve the manuscript.
We believe that all the reviewers and the editors would be happy to accept our manuscript at your well recognized journal.
Best regards,
Bat-Erdene PUREV-ERDENE

Reviewer 2 Report
Dear Reviewers,
Thanks for your feedbacks and corrections you did but please add the answer from comment 1 in the methodology to make clear why you have not used oder pozzolanic materials in this study.
The following literatures better to be cited in line 432 in front of "C-S-H" (https://doi.org/10.1021/acs.jpcc.8b11920) and "Portlandite" (https://doi.org/10.1016/j.apsusc.2019.144296), to make clear the name of CSH phases and their properties for readers and also morphology of portlandite from simulations and experimental research.
In conclusion, it will be accepted from my side and everything is clear for publishing.
Best regards,
Author Response
Rebuttal letter to the reviewer’s comments
Reviewer 2
Comments 1.
The following literatures better to be cited in line 432 in front of "C-S-H" (https://doi.org/10.1021/acs.jpcc.8b11920) and "Portlandite" (https://doi.org/10.1016/j.apsusc.2019.144296), to make clear the name of CSH phases and their properties for readers and also morphology of portlandite from simulations and experimental research.
In conclusion, it will be accepted from my side and everything is clear for publishing.
Answer: Thank you for accepting our manuscript for publication. We also thank you for handling the paper and reviewing it, which helped us to improve the manuscript. We have added citations as you suggested. As follows:
[42] X. Chen et al., “Morphology prediction of portlandite: Atomistic simulations and experimental research,” Appl Surf Sci, vol. 502, Feb. 2020, doi: 10.1016/j.apsusc.2019.144296.
[43] G. Xian, Z. Liu, Z. Wang, and X. Zhou, “Study on the Performance and Mechanisms of High-Performance Foamed Concrete,” Materials, vol. 15, no. 22, p. 7894, Nov. 2022, doi: 10.3390/ma15227894.

Reviewer 3 Report
The authors have made neccessary changes. Therefore manuscript can be accepted as it is.
Author Response
Reviewer 3
Comments:
The authors have made necessary changes. Therefore manuscript can be accepted as it is.
Answer: Thank you for accepting our manuscript for publication. We also thank you for handling the paper and reviewing it, which helped us to improve the manuscript.
